# A Survey on Knowledge, Attitude, and Practices of Large-Animal Farmers towards Antimicrobial Use, Resistance, and Residues in Mymensingh Division of Bangladesh

**DOI:** 10.3390/antibiotics11040442

**Published:** 2022-03-24

**Authors:** Md. Tarek Hossain, Kazi Rafiq, Md. Zahorul Islam, Sharmin Chowdhury, Purba Islam, Ziaul Haque, Mohammed Abdus Samad, Aminatu Abubakar Sani, Most. Rifat Ara Ferdous, Md. Rafiqul Islam, Nurnabi Ahmed, Md. Ismail Hossen, A. K. M. Khasruzzman, Mohammod Kamruj Jaman Bhuiyan, Muhammad Tofazzal Hossain

**Affiliations:** 1Department of Pharmacology, Bangladesh Agricultural University, Mymensingh 2202, Bangladesh; drtarek94@gmail.com (M.T.H.); drzahorul@bau.edu.bd (M.Z.I.); purba.islam@bau.edu.bd (P.I.); aminatu.sani@udusok.edu.ng (A.A.S.); rafkakoly.24@gmail.com (M.R.A.F.); 2Department of Pathology and Parasitology, Faculty of Veterinary Medicine, Chattogram Veterinary and Animal Sciences University, Khulshi, Chattogram 4225, Bangladesh; sharminchowdhury77@gmail.com; 3Department of Anatomy and Histology, Bangladesh Agricultural University, Mymensingh 2202, Bangladesh; zhaqueah80@bau.edu.bd; 4Antimicrobial Resistance Action Center, Animal Health Research Division, Bangladesh Livestock Research Institute, Savar, Dhaka 1341, Bangladesh; samad_blri@yahoo.co.nz; 5Livestock Division, Bangladesh Agricultural Research Council (BARC), Farmgate, Dhaka 1215, Bangladesh; mrislam210@hotmail.com (M.R.I.); ismailvet07@gmail.com (M.I.H.); 6Department of Parasitology, Bangladesh Agricultural University, Mymensingh 2202, Bangladesh; nurnabiahmed07@gmail.com; 7Department of Microbiology and Hygiene, Bangladesh Agriculture University, Mymensingh 2202, Bangladesh; dr.khasru@yahoo.com (A.K.M.K.); tofazzalmh@bau.edu.bd (M.T.H.); 8Department of Agricultural and Applied Statistics, Bangladesh Agricultural University, Mymensingh 2202, Bangladesh; mkjbhuiyan@bau.edu.bd

**Keywords:** large-animal farmers, survey, knowledge, attitude, and practices (KAP), antimicrobial use (AMU), antimicrobial resistance (AMR), antimicrobial residue (AR)

## Abstract

The widespread and indiscriminate use of antimicrobials in food animals is a key contributor to antimicrobial resistance and antimicrobial residue, which have become a growing public and animal health concern in developing countries such as Bangladesh. This study was aimed to assess the knowledge, attitude, and practices (KAP) of large-animal farmers towards antimicrobial use (AMU), antimicrobial resistance (AMR), and antimicrobial residue (AR) with their correlation. A cross-sectional survey was conducted with a structured and pretested questionnaire in the Mymensingh division of Bangladesh. A total of 212 large-animal farmers (dairy, beef fattening, buffalo, sheep, and goat farmers) were surveyed. Results showed that most of the farmers are male (85.8%) and belong to the 18–30 age group (37.3%). About 20.3% had no formal education, and nearly half of the participants (48.1%) received training regarding antibiotic use and resistance. Penicillin is the most common class of antibiotic used (61.8%) in the study area, followed by other antimicrobials. Only 37.7% of the farmers used antimicrobials on the recommendation of their veterinarian. Overall, 41.5%, 42.5%, and 21.7% of farmers possess adequate knowledge and a satisfactory attitude and perform desirable practices, respectively. Farmers in the 31–40 age group have adequate knowledge, attitude, and ability to implement desired practices compared to farmers in the 18–30 age group. Farmers having a graduate or post-graduate degree scored better in relation to knowledge, attitude, and practice than other farmers. Analysis revealed that farmers who received training on AMU and AMR had 10.014 times (OR = 10.014, 95% CIs: 5.252–19.094), 9.409 times (OR = 9.409, 95% CIs: 4.972–17.806), and 25.994 times (OR = 25.994, 95% CIs: 7.73–87.414) better knowledge, attitude, and performance, respectively, compared to their counterparts. A significant proportion of farmers (97.2%) dispose of leftover antibiotics inappropriately. The findings of the present study will be used to intervene in the education and training of the farmers, which will help to limit the indiscriminate and irrational use of antimicrobials, leading to reducing the chances of developing AMR.

## 1. Introduction

Antimicrobials are the drugs of choice for the treatment of infections in both humans and animals [1] and exhibit a variety of chemical and biological properties. In addition, antimicrobials are used for chemotherapeutic and prophylactic purposes and also to promote growth and improve feed efficiency in animals [2]. After the invention of antimicrobials, they saved millions of human and animal lives and improved their lifestyle [3]. Antimicrobial use (AMU) in the livestock sector is growing day by day due to the livestock revolution taking place, especially in middle- and low-income countries, due to the increasing demand of animal source food and livestock farmers also wanting quicker growth to maximize their profit [4].

Irrational AMU in animal production is thought to be a key contributor to antimicrobial resistance (AMR) despite the fact that the development of AMR is a complex multifactorial process [5,6]. AMU in livestock may affect AMR in humans as resistant bacteria that colonize in animals, developed by the selective evolutionary pressure of livestock AMU, can be transmitted to humans through the consumption of animal products, direct contact, and environmental exposure [7]. The emergence of resistant bacteria has been linked to the inappropriate use of antimicrobials in humans and animals [8]. The use of antimicrobials, regardless of their form or necessity, and improper dosing with too little, for too short a period, or the use of the wrong one can accelerate the AMR [9]. It is alarming that, due to the use of millions of tons of antimicrobials over the past decades, most of the disease-causing bacteria have become resistant to antimicrobials [10]. 

Currently, antimicrobial resistance has become a major public and animal health problem that threatens the effective prevention and treatment of a wide range of infections caused by bacteria [9]. AMR enhances the risk of morbidity, mortality, disease burden, longer duration of hospitalization, increased hospital costs, reduced livelihoods, and increased use of alternative drugs [11,12]. AMR has a direct, negative influence on the productivity of livestock, which may hamper safe food production [13]. 

Inappropriate use of antimicrobials and non-observance of withdrawal periods may result in the deposition of antibiotic residue in animal tissue [14,15]. Antibiotic residues that are deposited in food of animal origin must not be permitted for human consumption [16]. Antibiotic residue may lead to potential health hazards both in humans and animals; these include immunopathological effects, autoimmunity, carcinogenicity, teratogenicity, mutagenicity, nephropathy, hepatotoxicity, reproductive disorders, bone marrow toxicity, allergies, etc. [2,17,18]. In addition, antibiotic residue also plays an important role in the development of antimicrobial-resistant pathogenic bacteria [19]. Human exposure to significant levels of antibiotic residues from animal products can aggravate immune responses and may have a negative impact on the microflora in the intestine, which leads to the development of antimicrobial-resistant bacteria [20]. Antimicrobial-resistant bacteria may also develop through environmental contamination by antimicrobial residues excreted through the feces and urine of animals [21]. Infected animals may also serve as a reservoir for resistant bacteria, which may enter into the food chain [22]. 

Although AMR is a global health concern, people in low- and middle-income countries bear the brunt of the consequences [13]. Bangladesh, as a developing country in Southeast Asia, is vulnerable to the spread of AMR [23]. The misuse or abuse of antimicrobials in the livestock sector of Bangladesh is mainly due to inadequate veterinary healthcare facilities and sanitary conditions, malpractices by informal veterinary healthcare providers, poor monitoring and regulator surveillance, a high occurrence of diseases, and famers’ lack of knowledge on AMU and AMR [24]. Antimicrobial resistance is now being monitored and regulated by governments and international organizations throughout the world [25,26]. Several high-income countries have introduced control programs to monitor antimicrobial resistance; but, in most low- and middle-income countries such surveillance systems are lacking. Many of these countries also lack a legislative framework that regulates the use of antimicrobials within the livestock sector [13]. 

In accordance with the Global Action Plan (GAP) guidelines of the WHO, Bangladesh has developed and adopted a National Action Plan (NAP) for the prevention and control of AMR in human, animal, and environmental sectors for the period of 2017–2022 [24]. Noticeably, there is no particular policy or guideline for antibiotic use in the livestock sector in Bangladesh [27]. In Bangladesh, regulations on AMU are not well implemented; veterinary drugs are not commonly prescribed by veterinarians and are easily accessible to farmers. Antimicrobial agent use in farm animals is influenced by a variety of causes and incentives that are poorly understood. In this regard, a previous study highlighted that better knowledge, attitudes, and practices can explore information on antibiotic use and resistance, which can help future interventions to prevent both antibiotic misuse and the development of antimicrobial resistance [28]. Therefore, information on knowledge, attitude, and practices regarding AMU and AMR in a specific area is crucial and needed to identify farmers’ risky behavior and factors associated with them as possible targets for intervention [29]. Mymensingh is one of the division in Bangladesh with the largest large-animal populations. Farmers are involved in the production of small ruminants, dairy, and beef, utilizing a variety of management approaches. To the best of our knowledge, there is little information available in Mymensingh division or in Bangladesh about antibiotic use by food-producing large-animals farmers’ or their perceptions on irrational antimicrobial use, AMR, and antimicrobial residue [30]. Therefore, this study was undertaken to assess the large-animal farmers’ knowledge, attitudes, and practices towards AMU, AMR, and antimicrobial residue, which would help to limit the development of AMR, minimize the irrational use of antimicrobials, and would definitely help policy makers for the development of a proper interventional program for practical and sustainable changes in behavior.

## 2. Results

### 2.1. Socio-Demographic Characteristics of the Respondents

We conducted 212 interviews for the current investigation in all four districts of Mymensingh Division, namely, Mymensingh, Sherpur, Jamalpur, and Netrokona.

The characteristics of the study interviewees are shown in Table 1. Out of the 212 interviews, most of the respondents were male (85.8%, n = 182) and most of them belonged to the 18–30 year age group (37.3%, n = 79). Nearly one-fifth of participants (20.3%, n = 43) had no formal education, and almost half of the participants (48.1%, n = 102) received training regarding antibiotic use and resistance. Most of the farmers had a dairy farm (45.3%, n = 96) while goat farming (22.6%, n = 48) was second most among the farmers. In terms of farm animal population size, most of the farmers had 6 to 10 animals (43.9%, n = 93) on their farms (Table 1).

### 2.2. Common Antibiotics Used in the Study Area

Several antimicrobials were used on large-animal farms in the study area. Antimicrobials were used on the farm either alone or in combination with other antimicrobials. In the study, the most common class of antimicrobials used in large-animal farms were penicillin (61.8%), oxytetracycline (55.7%), sulfa drug (55.7%), and streptomycin (54.7%) followed by ciprofloxacin (51.9%), gentamicin (43.1%), and ceftriaxone (34.9%) (Figure 1).

### 2.3. Knowledge of Large-Animal Farmers on AMU, AMR, and Antimicrobial Residue

As shown in Figure 2, we asked 11 questions to assess respondents’ knowledge regarding AMU and AMR. Most of the respondents said that they had heard about antibiotics (96.7%) and antimicrobial resistance (71.2%). The large-animal farmers were more likely to say, “antimicrobial resistance causes treatment failure and poor response to treatment” (66.5%) when they were asked about antibiotic resistance. A larger proportion of the respondents (72.6%) did not know what actually antibiotics do or whether they work against bacteria or act against other organisms such as virus, fungus, and others. A sizeable proportion of farmers agreed that an incomplete antibiotic course may lead to antibiotic resistance (52.8%) and an overdose/low-dose course may lead to antibiotic resistance (42.5%). About 63.7% farmers answered yes when they were asked if they had heard about antibiotic residue. Interestingly, when they were asked what is antibiotic residue, more than half of the total (54.2%) described antibiotic residue as an accumulation of antibiotics in the human body through the ingestion of meat and milk during antibiotic treatment, the accumulation of antibiotics in the animal body, and/or both. About 60.8% of the farmers had heard about a withdrawal period of antibiotics and had some idea about a shelf-life/expiry date of antibiotics (84.9%). When they were asked if they had any knowledge about biosecurity, close to half of the respondent (46.2%) answered they did.

### 2.4. Attitudes of Large-Animal Farmers on AMU, AMR, and Antimicrobial Residue

We asked nine questions to the large-animal farmers to assess their attitudes towards AMU and AMR. The results are shown in Figure 3. Most of the farmers (79.7%) used the same antibiotics to prevent any specific disease regularly. Most of the farmers (70.8%) answered “negative” to the question, “Can antimicrobials be used to treat any kind of disease in animals?”. Similarly, a negative answer was found among farmers (70.3%) to the question, “Do you agree to sell animal products or slaughter animals during antimicrobial treatment or without maintaining a withdrawal period in order to reduce the cost of treatment?”. About 59.9% of farmers said “yes” when we asked them, “Do you stop antimicrobial treatment once animals feel better?”, which is considered bad for antimicrobial treatment. In our survey, it was reported that 50.9% of farmers thought the “use of antimicrobials may be reduced by maintaining proper biosecurity and vaccination”. About 54.2% of farmers agreed that antibiotics should be prescribed only by a veterinarian. Surprisingly, 78.3% gave a negative answer to the question, “Do you agree to alter the doses without consulting the prescribers to get a better response?”.

### 2.5. Practices of Large-Animal Farmers on AMU, AMR, and Antimicrobial Residue

We asked 10 questions to assess respondents’ practices regarding AMU and AMR. The results are shown in Figure 4. When we asked them “Who recommended you antibiotics?”, about 37.7% of farmers answered a veterinarian. Similarly, only 37.3% of farmers kept records of antimicrobial treatment. We found 64.2% of farmers completed an antibiotic course the last time. It was a matter of great regret that we found only 25.5% of the farmers followed a withdrawal period after the use of antibiotics. The majority of the farmers (98.1%) reported that they did not add antibiotics during self-feed processing, depicting a good practice. More than half of the farmers (51.9%) reported that they followed the exact prescription of a veterinarian during the purchasing of an antibiotic. Two-thirds of the total population size (68.9%) reported that they consumed or sold animal products or slaughtered animals during antimicrobial treatment or without maintaining withdrawal, depicting a bad practice.

### 2.6. Differences in Large-Animal Farmers’ Knowledge, Attitudes, and Practices

Principle factor analysis was performed to show the significant factors between the demographic variables and knowledge theme. The results are demonstrated in Table 2, showing that a farmer’s sex (*p* = 0.003), level of education (*p* < 0.001), training program regarding antimicrobial use and resistance from any institutions (*p* < 0.001), and their farm population size (*p* < 0.001) were the significant factors influencing the knowledge theme. The analysis revealed that a farmer’s sex (*p* < 0.001), level of education (*p* < 0.001), training received (*p* < 0.001), type of farm the farmer owned (*p* = 0.028), and farm population size (*p* < 0.001) were the significant factors affecting their attitudes. The analysis also revealed that a farmer’s age (*p* = 0.044), sex (*p* < 0.001), education (*p* < 0.001), training (*p* < 0.001), and farm population size were the significant factors influencing their practice (Table 2).

### 2.7. Differences in Respondents’ Knowledge, Attitudes, and Practices

The adjusted logistic regression analysis output on the farmers’ demographic variables and their levels of knowledge, attitudes, and practices is presented in Table 3. The results demonstrated that males had 4.192 times the odds of having a proper knowledge of AMU and AMR (OR = 4.192, 95% CIs = 1.53–11.43) compared with females. In these multivariable logistic regression predictors’ models, it was revealed that farmers aged 31 to 40 years were 1.281 times more likely to have adequate knowledge of AMU and AMR (OR = 1.281, 95% CIs: 0.672–2.443) compared to the 18–30-year-old group. The analysis also revealed that farmers educated at PSC, JSC, SSC, HSC, graduate, and masters levels were found to be 1.123 times (OR = 1.123, 95% CIs: 0.288–4.369), 1.987 (OR = 1.987, 95% CIs: 0.47–8.394), 3.231 (OR = 3.231, 95% CIs: 0.835–12.496), 2.816 times (OR = 2.816, 95% CIs: 0.799–9.928), 2.045 times (OR = 2.045, 95% CIs: 0.43–9.717), and 2.513 times (OR = 2.513, 95% CIs: 0.401–15.746) more likely to have adequate knowledge of AMU and AMR, respectively, compared to illiterate farmers. Further, the farmers who received training regarding antibiotic use and resistance from any institution had 10.014 times the odds of having ‘correct’ knowledge of AMU and AMR compared with their counterparts (OR = 10.014, 95% CIs: 5.252–19.094). Regarding a farmer’s farm population, farmers who had 6–10, 11–20, and >20 animals on their farms had 1.84 times (OR = 1.84, 95% CIs: 0.569–5.95), 2.515 times (OR = 2.515, 95% CIs: 0.623–10.157), and 23.147 times (OR = 23.147, 95% CIs: 4.214–127.131) the odds of having adequate knowledge of AMU and AMR than those who had 3–5 animals on their farm. There was no significant variation found for the other variables. 

### 2.8. Relationship among Knowledge, Attitudes, and Practices of AMU and AMR

In terms of attitudes, the results showed that males had 5.823 times the odds of having a ‘better’ attitude towards AMU and AMR (OR = 5.823, 95% CIs: 1.954–17.356) compared with female farmers. They further revealed that the respondents who belonged to the age group 31–40 had a ‘better’ attitude (OR = 1.35, 95% CIs: 0.707–2.578) compared with the farmers who were in the 18–30 age group. Furthermore, farmers educated at PSC, JSC, SSC, HSC, graduate, and masters levels were found to be 1.25 times (OR = 1.25, 95% CIs: o.441–3.54), 3.646 times (OR = 3.646, 95% CIs: 1.169–11.373), 3.603 times (OR = 3.603, 95% CIs: 1.268–10.236), 6.836 times (OR = 6.836, 95% CIs: 2.535–18.43), 10.208 times (OR = 10.208, 95% CIs: 2.994–34.807), and 39.379 times (OR = 39.379, 95% CIs: 4.345–356.834) more likely to have better attitudes regarding AMU and AMR, respectively, compared to illiterate farmers. Further, the farmers who received training regarding antibiotic use and resistance from any institution had 9.844 times the odds of having better attitudes on AMU and AMR compared with their counterparts (OR = 9.844, 95% CIs: 5.190–18.670). Regarding a farmer’s farm population, farmers who had 6–10, 11–20, and >20 animals on their farms had 2.313 times (OR = 2.313, 95% CIs: 0.962–5.561), 5.946 times (OR = 5.946, 95% CIs: 2.292–15.430), and 55.5 times (OR = 55.5, 95% CIs: 10.848–283.950) the odds of having better attitudes regarding AMU and AMR than those who had 3–5 animals on their farms. There was no significant variation found for the other variables. 

The analysis of AMU and AMR practices showed that males had 1.127 times the odds of having a ‘better’ practice towards AMU and AMR (OR = 1.127, 95% CIs: 0.431–2.946) compared with female farmers. They further revealed that the respondents who belonged to the age group 31–40, 41–50 years, and >50 years had a ‘better’ practice (OR = 2.962, 95% CIs: 1.32–6.645; OR = 1.686, 95% CIs: 0.636–4.466; and OR = 1.091, 95% CIs: 0.274–4.348, respectively) compared with the farmers who were in the 18–30 age group. Furthermore, farmers educated at PSC, JSC, SSC, HSC, graduate and masters levels were found to be 2.562 times (OR = 2.562, 95% CIs: 0.470–13.980), 3.237 (OR = 3.237, 95% CIs: 0.499–21.003), 5.979 times (OR = 5.979, 95% CIs: 1.148–31.141), 8.483 (OR = 8.483, 95% CIs: 1.764–40.081), 20.5 times (OR = 20.500, 95% CIs: 3.866–108.698), and 47.833 times (OR = 47.833, 95% CIs: 6.730–339.760) more likely to have better practices regarding AMU and AMR, respectively, compared to illiterate farmers. Further, the farmers who received training regarding antibiotic use and resistance from any institution had 25.994 times the odds of having better practices regarding AMU and AMR compared with their counterparts (OR = 25.994, 95% CIs: 7.730–87.414). Regarding a farmer’s farm population, farmers who had 6–10, 11–20, and > 20 animals on their farms had 1.75 times (OR = 1.750, 95% CIs: 0.349–8.786), 14.086 times (OR = 14.086, 95% CIs: 3.046–65.134), and 48.375 times (OR =48.375, 95% CIs: 9.344–250.451) the odds of having better practices regarding AMU and AMR than those who had 3–5 animals on their farms.

There was no significant variation found for the other variables in the present study.

According to Spearman’s rank-order correlation, each pair of respondents’ knowledge, attitude, and practice scores had a positive relationship (*p* ≤ 0.001). Knowledge–attitudes, knowledge–practices, and attitudes–practices all showed a reasonable correlation [31], as shown in Table 4.

## 3. Discussion

AMR has become a global issue that threatens both human and animal health [32]. Bangladesh is a key contributor to AMR due to its poor healthcare standards and antibiotic abuse and overuse [33]. Stakeholders in the livestock sector, including farmers who operate as end users, must participate in reducing the danger of antibiotic resistance [32]. The monitoring of antimicrobial use is suggested by international organizations because it provides helpful information for policy creation to reduce AMR concerns [26]. The current study assessed large-animal farmers’ knowledge, attitudes, and practices (KAP) about AMU, AMR, antibiotic residues, and biosecurity management of the farm. We found that respondents’ age, sex, education, and farm type and size all had an impact on their KAP toward AMU and AMR. To the best of our knowledge, this is the first study among large-animal farmers in a selected area of Bangladesh. The research offered baseline evidence concerning KAPs of large-animal farmers regarding AMU, which would definitely help to design interventions to minimize antimicrobial abuse or overuse for combating AMR. 

Antibiotics were used by all of the farmers in our survey. This study found that antimicrobial drugs were widely used in livestock production by the large-animal farmers in the study locations to limit the source of infection in farms as a result of inadequate management. These findings correspond to recently published reports [32,34]. Our research revealed that antimicrobial usage can differ significantly between and within countries, species, production systems, and individual farms; these factors are in agreement with a previous report [35]. This present study also observed that there was some heterogeneity in drug choice and the number of respondents who had used antimicrobials. In addition, the present study showed that several antimicrobials were used to treat various large-animal diseases, either alone or in combination with other antimicrobials. The most common antibiotic used by the farmers was penicillin, which is similar to a previous study among dairy farmers in the United Kingdom [36]. Common antimicrobials such as beta lactum, tetracycline, sulfonamides, aminoglycosides, macrolides, and cephalosporin were used in the study area, which is consistent with previous studies, and they reported that these groups of antimicrobials are widely used in large-animal production [37,38,39]. The World Health Organization considers the majority of these antimicrobials to be either crucial (amoxicillin, gentamicin, and ampicillin) or highly important (sulfonamides, doxycycline, and oxytetracycline) for humans [40]. As a result, their residues in dairy products are conveyed to people through consumption. Human consumption of antimicrobial-contaminated milk and meat could lead to teratogenic effects, reduction in reproductive performance, allergies, acute toxicity, carcinogenicity, and the emergence of AMR bacteria, leading to the risk of AMR development [2,41].

### 3.1. The Knowledge of Large-Animal Farmers Regarding AMU and AMR

In this present study, 41.5% of large-animal farmers had adequate knowledge about AMU and AMR, which is lower than previous studies recorded in Malaysia, Algeria, and the UK [39,42,43]. In contrast, 30% of livestock keepers in Ethiopia, 10% of animal producers in Turkey, and 7.5% of livestock and aquaculture owners in Vietnam were aware of the proper AMU and had a good understanding of AMR formation [28,44,45]. One of the major contributors to the rise of AMR is antibiotic misuse, which is linked to an antimicrobial knowledge gap [46,47,48]. The results of the present study showed that 96.7% of farmers heard about antibiotics but only 24.7% of farmers answered correctly when they were asked about ‘what antibiotics do’. These findings are consistent with a previous study in Cameroon [32]. On the other hand, the result of the present study revealed that 56.7% and 63.3% of farmers correctly knew about antimicrobial resistance and antimicrobial residue, respectively. These findings are more or less similar to a previous report in Bangladesh [46]. Differences in the knowledge of antibiotics and antimicrobial resistance and residues found in the current study may be due to exposure to communication and mass media sources. These findings may reflect the information provided by the sources to the farmers, which may focus on communicating the risk connected with antimicrobial resistance and antimicrobial residue, and provide less information about antimicrobials in general. Although a high percentage of large-animal producers are aware of antimicrobial resistance and antimicrobial residue compared to the abovementioned data from developing countries, there is still a need for additional education on AMR and AMU. Another finding of the current study was that most of the farmers (84.9%) knew that antibiotics have side effects. This finding is higher than the findings reported in previous studies, but this outcome was in line with what had been found in prior studies of Bangladesh [34,45,46]. The majority of the farmers (60.8%) in the current study had knowledge about the withdrawal period of antimicrobial drugs. Nearly half of the farmers (47.2%) did not know that an incomplete antibiotic course may lead to antibiotic resistance. Another finding of the current study was that more than half of the farmers (57.5%) did not believe low dose or overdose of antibiotic courses may lead to antibiotic resistance. These findings are consistent with a previous study in Ethiopia [34]. We believe that all of these misconceptions are linked to an increase in antimicrobial-resistant microorganisms. In addition, one of the key causes of the widespread overuse of antimicrobials in farms is a lack of understanding of on-farm management, including biosecurity measures. The results of the present study showed that less than half of the farmers (46.2%) had knowledge about biosecurity. More than half of the farmers (50.9%) agreed that maintaining proper biosecurity, vaccination, and good management practices could reduce the use of antimicrobials on farms. In this regard, a previous study’s finding was higher than the current study’s report [49]. On the other hand, 96% of livestock farmers in Vietnam and 82% of livestock keepers in Ethiopia felt that an alternative to AMU, such as immunization, could be beneficial in lowering AMR generation [28,44]. On the other hand, in Italy, 47% of turkey farmers and 78% of rabbit farmers believed that genetic improvement of the breeds could help them reduce wasteful AMU [50]. The above discussion indicates that reduction of AMU in large-animal farms could be possible by improving good management practices.

### 3.2. The Attitudes of Large-Animal Farmers Regarding AMU and AMR

The current study revealed that 42.5% of the respondents showed a desirable attitude towards AMU and AMR. This finding was lower than in previous studies [44,51]. On the other hand, a study showed that only 14.7% of the farmers had a desirable attitude towards AMU and AMR [49]. Most of the farmers (70.8%) believed that antimicrobials could be used to treat any kind of disease. The findings of the current study are in agreement with a study of poultry farmers in Bangladesh but are higher than a study reported in Turkey [45,46]. A previous study reported that more than half of the farmers (54.2%) believed that antibiotics should only be prescribed by veterinarians [49]. This finding is in agreement with the findings of the current study. Encouragingly, almost all farmers (98.1%) in this current study showed a desirable attitude towards not using antibiotics as growth promoters. These findings are in contrast with some previous studies [34,46]. This result indicates that farmers in the study area are well aware of the detrimental effects of antibiotics as a growth promoter and changed their attitude towards the AMU. According to previous surveys, nearly half of the respondents showed an undesirable attitude toward the fact that they raised the antibiotic dose and frequency as long as the animals showed no indications of recovery [34,45]. These findings are much higher than the current study, of which 21.7% of the farmers said that they altered the doses without consulting the prescribers to get a better response. Our finding is in line with a study reported in Ethiopia [44]. A large proportion of farmers (40.1%) showed an undesirable attitude by stopping antimicrobial treatment once animals felt better. This finding is lower than some previous studies [34,45]. In the present study, we found that a major portion of farmers (68.9%) did not read the prospectus before antimicrobial treatment. This finding is higher than some studies reported from Turkey and Ethiopia [34,45]. 

### 3.3. The Practices of Large-Animal Farmers Regarding AMU and AMR

Large-animal farmers’ practices regarding AMU and AMR in this study were inappropriate and were in line with previous findings [44,49]. On the other hand, some previous research stated that 54% of the farmers performed appropriate practices, which was higher than our results. We found that 37.7% of the farmers took antibiotic prescriptions only from a veterinarian, which is lower than several previous studies in which livestock producers sought veterinarian guidance before using antimicrobials for any animal production purpose [28,39,44,45,49]. Because of the inadequate governmental animal healthcare system in Bangladesh, farm owners rely on unqualified and informal healthcare practitioners to treat their animals. Diagnosis of diseases by the owners or other farmers also plays an important role for not following veterinary guidance. As a result, arbitrarily given antibiotics and easy access to them led to their misuse, abuse, and suboptimal or overuse in farms [52]. Access to antimicrobials without a prescription and fragmented governance of AMU in animal production are the main drivers of AMR generation, as described by many researchers [53]. Although Bangladesh has lately taken a number of steps to reduce antibiotic usage in order to address antimicrobial resistance [52], progress on animal health remains slow and insufficient. As a result, animal owners can still obtain antibiotics without a prescription from veterinary pharmacies. The results of the study also revealed that only 25.5% of farmers follow the withdrawal period of antibiotics. This finding is lower than some studies reported for poultry farmers in Bangladesh but higher than a study in Nigeria [30,54]. Violations of product label withdrawal periods, such as those seen in this study, have been recorded in other investigations as well [37,55]. On the contrary, according to research conducted in Vietnam [28], over 90% of chicken and pig producers followed the withdrawal period and stopped using antibiotics before selling their products. Antimicrobial residues were found in dairy products due to some farmers’ continued noncompliance with antimicrobial withdrawal periods. As a result, the emergence of novel infections harboring AMR genes was boosted [56]. A study from Lebanon stated that dairy farmers’ noncompliance may also be due to a fear of financial loss if milk is discarded during the withdrawal period [57,58,59]. When the farmers who did not follow a withdrawal period of antimicrobials in the current study were asked about the causes of not following a withdrawal period, more than half of them (52.5%) said that they did not know about it, which was why they did not follow it, and 40.5% answered in favor of reducing economic loss. This finding suggests that there is a need to improve the knowledge of farmers by giving proper training or by providing information through communication or mass media. Another finding of this study indicated that 37.3% of farmers kept proper records about the use of antibiotics, which is consistent with a previous study in the USA [60]. According to the findings of this study, there has been an increase in the number of large-animal producers keeping records on antimicrobial usage in recent years. This rise in record keeping could be due to farmers’ increasing understanding of the need to keep records. Farmers reported that prescribers did not inform them about the dangers of antibiotic residue in large-animal products or that antibiotic abuse poses a major health risk to humans. Encouragingly, almost all farmers (98.1%) did not add antimicrobials to the animal feed, which indicates that large-animal farmers in the current study were using the appropriate practice. A study from Ghana reported that a large portion of farmers (63%) had the trend of not completing an antibiotic course [61]. This finding is higher than the current study’s result (35.8%). The causes of this finding may be due to monetary problems of the farmers, animals feeling better sooner, antibiotics’ tablets or injections running out, or the disease not curing or stopping according to an informal veterinary healthcare provider’s advice. The majority of the respondents (68.9%) showed inappropriate practice by not reading the prospectus of the antimicrobials. This finding is lower than the result of a study reported in Ethiopia but in line with a study in Turkey [44,45]. Two-thirds of the farmers (33.5%) stored medicine in the right place, showing an appropriate practice of AMU in our study. This finding is higher than previous studies in Turkey and Bangladesh [30,45]. Almost all of the respondents (97.2%) performed inappropriate practices with leftover antibiotics. Nearly half of the farmers (47.1%) replied that they kept the leftover antibiotics for further use. The same portion of farmers answered that they throw it in the garbage, and 2.8% said that they give it to other farmers. Only 2.8% of the farmers showed appropriate practices, with the answer that they either bury them in the ground or burn leftover antibiotics.

### 3.4. Association of Socio-Demographic Data with KAP of Large-Animal Farmers

The knowledge gap in AMR development emanating from livestock industries in resource-constrained environments has been widely discussed in previous studies [48,62,63]. Demographic parameters such as age, sex, years of experience, etc. of the respondents had a substantial impact on knowledge, attitudes, and practices [29]. The present study’s findings mirrored those of previous research. Age, sex, level of education, and training, as well as farm type and size, were revealed to be important predictors of farmers’ KAP regarding AMU and AMR in large-animal farms.

According to the current study, males had 4.192 times the odds of having proper knowledge of AMU and AMR, 5.823 times the odds of having ‘desirable’ attitudes, and had 1.127 times the odds of having ‘better’ practice towards AMU and AMR compared to female farmers. Findings of several studies [34,44,64] are consistent with the current survey results. The present study shows that female farmers have inadequate knowledge and undesirable attitudes compared to male farmers, but females showed better practices than males. Male farmers might have been more knowledgeable about AMU, AMR, and antibiotic residues than females because of their exposure to meetings, training, and media in the research location. Female participation in meetings, training, and other activities is uncommon in the research area. The results of the current study showed that only 26.7% (n = 30) of female farmers received training regarding livestock farming; on the other hand, 51.6% (n = 182) of male farmers received training on livestock farming from any kind of institution. Interestingly, another study in Vietnam reported that respondents, in spite of having superior knowledge and attitudes towards AMR, performed inappropriate AMU practices [28]. These current data suggest the need for improvement of knowledge of female farmers through educational campaigns, seminars, and participation in training programs regarding AMU and AMR.

Similar to prior surveys [32,45,54], the current study found that the age of the farmers has a significant relationship with knowledge, attitudes, and practices of AMU and AMR, and the intensity of this association differed from nation to nation. The results of the current study showed farmers aged 31 to 40 years had better knowledge of AMU and AMR, showed desirable attitudes, and performed better practices, compared to the 18–30-year-old group of farmers. These findings showed a similarity with a previous study in Bangladesh [65]. 

To combat against AMR, human behavior and educational level are crucial [66]. In addition, in order to use antimicrobials effectively, farmers must have a high level of education and adopt certain behaviors [29]. A farmer’s educational status is substantially linked (*p* < 0.05) with his/her knowledge, attitudes, and practices regarding AMU and AMR [49]. The results of the current study showed that farmers who finished their education up to the graduate level showed good KAP responses towards AMU and AMR. As with our current finding, a similar result was reported in a study in Turkey, which demonstrated that farmers holding graduate and post-graduate degrees were extremely educated about the usage of antibiotics compared to farmers with only a high school or primary education [45]. Farmers with a higher level of education may be more aware and have more access to veterinary services, farm management, and biosecurity measures, as well as a better understanding of antimicrobials’ use and their withdrawal periods [67]. 

In this study, farmers with inadequate knowledge scores were linked to a higher number of respondents who had no training (51.5%). The farmers who received training regarding antimicrobial use and resistance from any institution had 10.014 times the odds of having adequate knowledge of AMU and AMR, 9.844 times the odds of having desirable attitudes, and 25.994 times the odds of having appropriate practices compared with their counterparts. According to a study in Cameroon, farmers with lower knowledge scores were more likely to be untrained in poultry farming [32]. A previous study from Bangladesh on drug and feed sellers reported that farmers who received training had appropriate practices regarding AMU and AMR, which reflects our current findings [65]. Therefore, our findings suggest that farmers should be required to participate in training programs more and more regarding AMU and AMR. A farmer’s training would have been an important instrument in establishing a baseline of knowledge of AMU, AMR, and antimicrobial residues. Educational campaign, seminars, and mass media communications should be organized to train large-animal farmers with the help of physicians and veterinarians, who are the most reliable sources of health information for them.

Farmers having larger population size farms are more likely to have a favorable attitude to limit AMU than their smaller counterparts [57,58,59]. A previous study from Bangladesh reported that small-scale poultry farmers possessed lower knowledge, attitude, and practice scores than large-scale poultry farmers. The results of the current study are in line with a previous study [57], which demonstrated that farmers who had a large farm population size (more than 10 animals) had more adequate knowledge, showed desirable attitudes, and were more appropriate in practices than small population size farms. Economic status of the farmers had a significant role on a farmer’s KAP towards AMU, AMR, and antibiotic residues. In developing countries, poverty has been identified as the primary cause of antimicrobial abuse [68]. A study from Bangladesh reported that small-scale farmers are usually poor; the total amount spent on animals by households looked to be disproportionate to their income. The high cost of veterinary medicine, cost of animal healthcare, feeding cost of animals, and loss of an animal might play a major role in undesirable attitudes and inappropriate practices of large-animal farmers towards AMU and AMR [52]. The data from the current study suggest that government support would benefit the large-animal sector in Bangladesh. If farmers were given a financial incentive, they would be more eager to minimize AMU and, thus, would help to reduce the development of AMR bacteria [69].

### 3.5. Limitations of the Study

This study has some limitations due to the nature of gathering data on human behavior through survey approaches. In this survey, a KAP questionnaire was used to collect data. A small number of the respondents were selected from each of the four districts in Mymensingh, Bangladesh, which may not reflect the actual status of KAP for large-animal farmers. Furthermore, because participants self-reported their attitudes and past behaviors, there is a chance of incorrect recall and social desirability or confirmation bias, which could skew the results. Another limitation of the study is KAP survey techniques may mistakenly lead participants to provide responses that they foresee the researcher viewing as appropriate or desirable. The cause-and-effect relationship between the predictor variables and the dependent binary variables (knowledge, attitude, and practice) of large-animal farmers may be influenced by the nature of this cross-sectional survey. The number of questions was also reduced in order to cut down on the time it took to complete the survey.

## 4. Materials and Methods

### 4.1. Study Location and Study Period

The study was conducted in all four districts (Mymensingh, Netrokona, Sherpur, and Jamalpur districts) in Mymensingh division of Bangladesh (Figure 5: left panel; shown in box as study_area). Considering four upazillas (a district’s lowest administrative boundary) from each district, a total of 16 upazillas were surveyed (Figure 5: right panel; shown as study_upazilla) within Mymensingh division. These upazillas were selected on the basis of data (highest density of large-animal farms) provided by the corresponding District Livestock Offices. The study was carried out for 6 months, from July 2019 to December 2019.

### 4.2. Study Design and Sampling

A cross-sectional KAP survey was conducted on large-animal farmers regarding AMU, AMR, and antimicrobial residue. Data were collected from 212 farmers (96 dairy, 64 goat, 32 beef fattening, 16 sheep, and 20 buffalo farmers). A farmer was described as someone who daily spends time on feeding, watering, rearing, or caring for animals on the farm, having direct or indirect contact with a farm, and obviously with an age ≥18 years. Farmers were selected on a random basis from a list obtained from the corresponding Upazilla Livestock Offices. From each upazilla, a total of 12 farmers were surveyed, covering 6 dairy farmers, 3 goat farmers, 2 beef fattening farmers, and 1 sheep farmer. However, buffalo farmers (n = 20) from 6 buffalo-concentrated upazillas (Melandah, Islampur, Modon, Atpara, Sherpur sadar, Trishal) of four districts were selected. The Raosoft sample volume calculation method [65] was used to determine the sample size on the basis of a 5% margin of error (population size was 20,000), 85% confidence level, and assumption of response distribution of 50% [65], after adding a 5% nonresponse rate. Based on this method, the minimum sample size was 206; the sample size for this study was 212.

### 4.3. Preparation of Questionnaire

A structured questionnaire was developed based on previous studies [28,29,30,36,39,43,44,65]. The questionnaire was made up of general, descriptive, close-ended, open-ended, and multiple choice questions to assess a farmer’s knowledge, attitude, practice, and management of a farm regarding AMU, AMR, and antimicrobial residue. The survey questionnaire was structured in four major parts. The first part consisted of information about demographic characteristics such as sex, age, education, type of farm, farm population size, years of experiences, and training received regarding AMU and AMR. In the second part, questions were asked to the farmers about knowledge of antimicrobials, AMU, AMR, and antimicrobial residue. A farmer’s attitude and practices regarding AMU, AMR, and antimicrobial residues were assessed in the third and fourth parts, respectively. Besides this, some additional questions were also asked to the farmers about common antibiotic uses on the farm, reasons for not completing a withdrawal period, what the farmers did with leftover antimicrobials, and how AMU affects the economy of their farm.

### 4.4. Questionnaire Administration

Both English and Bangla versions of the questionnaire were developed to collect information from the farmers (Appendix A). Paraphrasing of the questionnaire was developed for easy communication with the respondents who did not have a formal education. The questionnaires were pretested before administration of the survey to the farmers to ensure question clarity, refinement, and timing accuracy. During pretesting and the main survey, one veterinarian and two trained enumerators were involved in administering the questionnaire and collecting data from farmers. Data were collected through face-to-face interviews. The research goals and the benefits as well as risks of involvement were explained to the participants prior to the survey. The participants were well informed that they could participate or withdraw at any time; we took their consent regarding the survey procedure before the interview. 

### 4.5. Data Management and Analysis

Data from the interviews were entered and cross-checked in a paper-based questionnaire. The information was then transferred to an MS Excel spreadsheet (Microsoft Excel 2018, Microsoft Corp, Redmond, WA, USA) for cleaning, processing, and analysis. We gathered information on various knowledge, attitudes, and practices related to AMU and AMR using a variety of closed-ended and open-ended questions. The correct (adequate/desirable/appropriate) response (‘yes’) was assigned a value of 1, while the incorrect (inadequate/undesirable/inappropriate) response (‘no’) was assigned a value of 0. The aggregate of each participant’s responses for that particular segment was tallied to see how they did overall in the knowledge, attitude, and practice areas. The data were examined using IBM SPSS Statistics (IBM Corp. Released 2017. IBM SPSS Statistics for Windows, Version 25.0. Armonk, NY: IBM Corp). The data were coded, and the internal consistency of the themes was assessed using Cronbach’s alpha test, with acceptable values of 0.814 for knowledge, 0.76 for attitude, and 0.71 for practice (0.89 when combining all the themes). The internal consistency of the data was good by the scale of Cronbach’s coefficient alpha value [70].

### 4.6. Pearson Chi-Square Test

The demographic and socioeconomic aspects affecting large-animal farmers’ management, knowledge, attitudes, and practices regarding AMU and AMR were studied individually. Age, sex, education, training, and a farmer’s farm size and farm type were all transformed into categorical variables before being analyzed. The association between a large-animal farmer’s knowledge, attitude, and practices with his/her demographic features was assessed using the Pearson chi-square test. The answers from the respondents regarding knowledge, attitude, and practices were categorized as “adequate” versus “inadequate”, “desirable” versus “undesirable”, and “satisfactory” versus “unsatisfactory”, respectively. For multiple logistic regression analysis, any explanatory variable linked with management, knowledge, attitude, and practices with a *p*-value of less than 0.20 was considered. Cramer’s phi-prime statistic was used to determine if explanatory variables were collinear. If Cramer’s phi-prime statistic was greater than 0.70, a pair of variables was called collinear [71].

### 4.7. Multivariable Logistic Regression Analyses

To explore the effects of demographic characteristics linked with AMU, AMR, and antimicrobial residues on the farmers’ knowledge, attitudes, behaviors, and farm management, we used the enter method for multiple logistic regression models. Variables such as sex, age, education, training, and farm size were categorized. The levels were for sex “male and female”; “illiterate, PSC, JSC, SSC, HSC, graduate and masters” for education; receiving “training or not” for training; and “the number of animals in a farm” for farm size. Multicollinearity among these potential explanatory variables was assessed by pair-wise Pearson correlation tests in IBM SPSS Statistics (IBM Corp. Released 2017. IBM SPSS Statistics for Windows, Version 25.0. Armonk, NY: IBM Corp). A pair of explanatory variables was considered collinear if Cramer’s phi-prime statistic was >0.70 [71]. The final multivariable models were automatically selected based on the lowest Akaike’s information criterion (AIC) value. Hosmer–Lemeshow goodness-of-fit tests were used to assess the overall model fit [72]. The results were reported as odds ratio (OR) with 95% confidence intervals (CIs) as the statistical significance criterion. The strength and direction of the association between responses to the management’s knowledge, attitudes, and practices questions were described using Spearman’s rank-order correlation coefficient [31].

## 5. Conclusions

The anticipated survey data uncovered the perspective of large-animal farmers in Mymensingh division of Bangladesh on knowledge, attitudes, and practices regarding AMU, AMR, and antibiotic residues. The findings of the study suggest that a significant number of large-animal farm owners/workers have inadequate knowledge, undesirable attitudes, and inappropriate antibiotic use practices towards AMU, AMR, and antibiotic residues. The present study data suggested that socio-demographic factors such as sex, age, training, farm population size, and particularly level of education have a significant impact on KAP towards AMU, AMR, and antibiotic residues. The findings of this study provided baseline evidence concerning the KAP of large-animal farmers, which would definitely help respective authorities to focus future interventions on Bangladesh smallholder livestock farming systems to reduce antimicrobial use and resistance. Regulation and control of the usage of veterinary medication and enacting strong antibiotic prescription legislation in Bangladesh to minimize widespread antimicrobial use are strongly recommended. Finally, it is recommended that AMU policies be developed and enforced in a way that is both practical and inclusive.

## Figures and Tables

**Figure 1 antibiotics-11-00442-f001:**
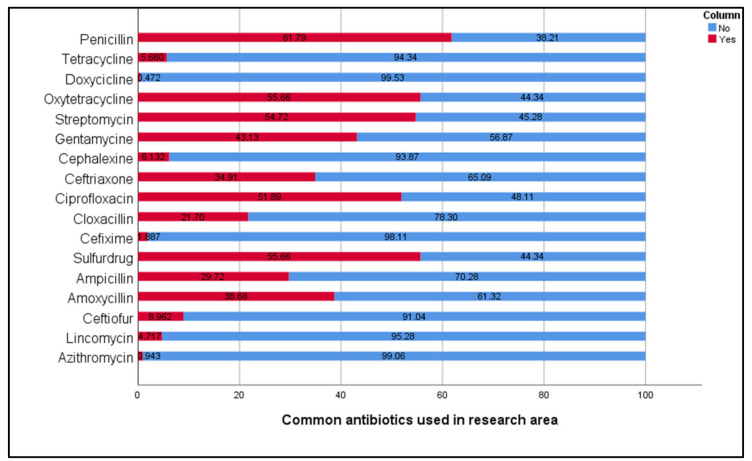
Common antimicrobials used on livestock in the study area (%).

**Figure 2 antibiotics-11-00442-f002:**
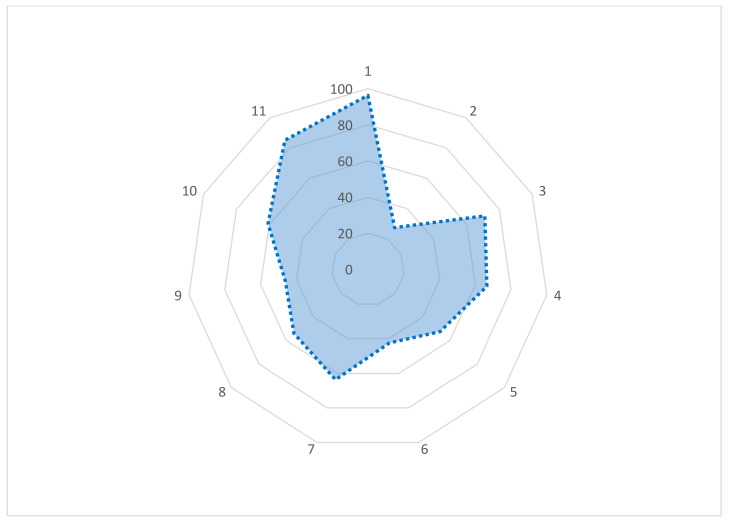
Radar chart of knowledge assessment of large-animal farmers’ answers to different questions in the survey questionnaire. (1) Have you heard about antibiotics? (Yes/No). (2) What do antibiotics do? (Act against bacteria/ act against virus/ act against fungus, others/act against all of the above/do not know). (3) Have you heard about antimicrobial resistance? (Yes/No). (4) What do you know about antibiotic resistance? (It causes treatment failure/it causes poor response to treatment/both/do not know/others). (5) Do you know an incomplete antibiotic course may lead to antibiotic resistance? (Yes/No). (6) Do you know an overdose/low-dose course may lead to antibiotic resistance? (Yes/No). (7) Have you heard about antibiotic residue? (Yes/No). (8) What is antibiotic residue? (Accumulation of antibiotics in the human body through the ingestion of meat and milk during antibiotic treatment/accumulation of antibiotics in the animal body/both/do not know). (9) Do you have any knowledge about biosecurity? (Have/do not have). (10) Have you heard about a withdrawal period of antibiotics? (Yes/No). (11) Do you know antimicrobials have some side effects? (Yes/No).

**Figure 3 antibiotics-11-00442-f003:**
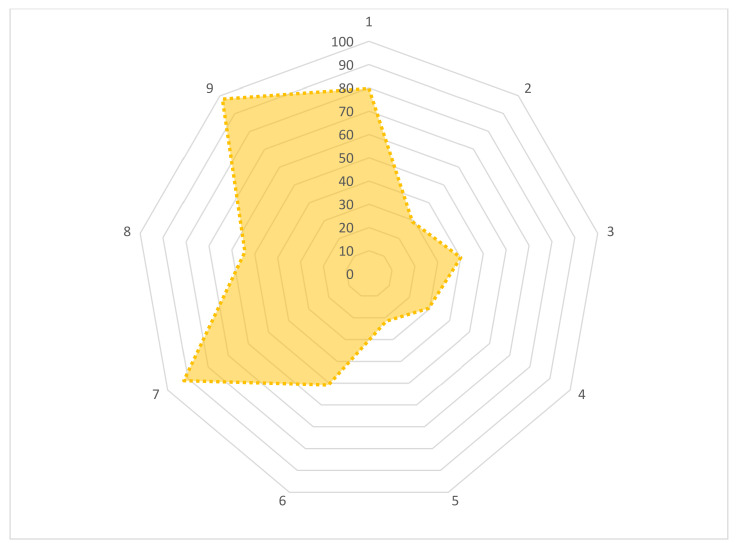
A radar chart depicts the distribution of desirable attitudes among large-animal farmers. (1) Do you use the same antibiotics to prevent any specific disease regularly? (Yes/No). (2) Can antimicrobials be used to treat any kind of disease in animals? (Yes/No). (3) Do you stop antimicrobial treatment once animals feel better? (Yes/No). (4) Do you agree to sell animal products or slaughter animals during antimicrobial treatment or without maintaining a withdrawal period in order to reduce the cost of treatment? (Agree/strongly agree/disagree). (5) Do you agree to alter the doses without consulting the prescribers to get a better response? (Yes/No). (6) Do you think the use of antimicrobials may be reduced by maintaining proper biosecurity, vaccination, and good management practices? (Yes/No). (7) Should antibiotics be used only when needed? (Agree/strongly agree/disagree). (8) Should antibiotics be prescribed only by veterinarians? (Agree/strongly agree/disagree). (9) Is the use of antibiotics as growth promoters necessary in livestock production? (Agree/strongly agree/disagree).

**Figure 4 antibiotics-11-00442-f004:**
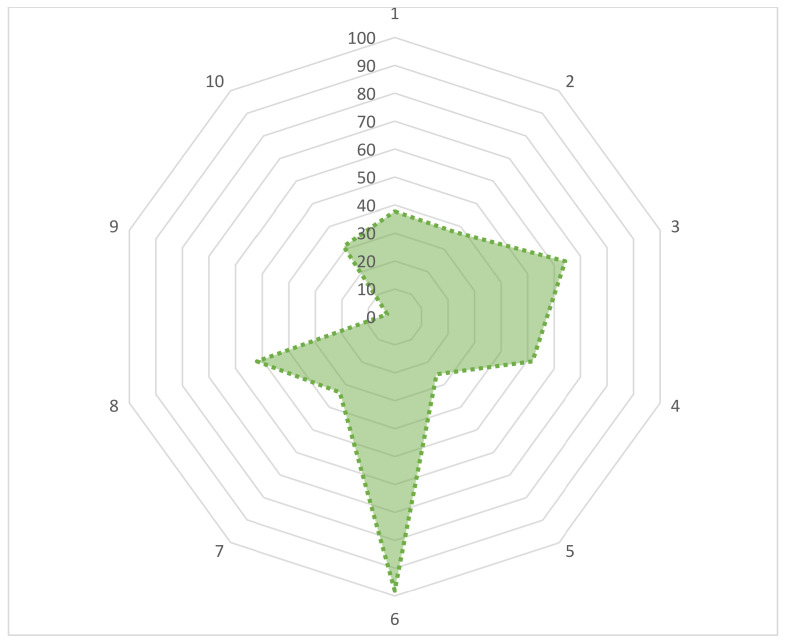
The distribution of appropriate practices among farmers for various questions in the study questionnaire. (1) Who recommended you antibiotics? (Veterinarian/other farmers/shopkeepers/representative of pharmaceutical company/veterinary paraprofessional/village doctor/ quack/self). (2) Do you keep a record of using antimicrobials? (Always/most frequently/sometimes/rarely/never/do not know). (3) Did you complete the antibiotic course the last time? (Yes/No). (4) Number of antibiotics used at a time on your farm? (Single/combined/both/do not know). (5) Withdrawal period follows? (Yes/No). (6) Do you add antibiotics to the feed of animals? (Yes/No). (7) Where do you store drugs? (Storeroom/refrigerator/shed/bedroom/others). (8) Do you follow the exact prescription of a veterinarian when purchasing the antibiotics? (Always/sometimes influenced by medicine seller/ others). (9) What do you do with leftover antibiotics? (Keep for further use/throw in the garbage/give them to other farmers for use/bury in the ground/burn). (10) Do you read the prospectus before using antimicrobials? (Yes/No).

**Figure 5 antibiotics-11-00442-f005:**
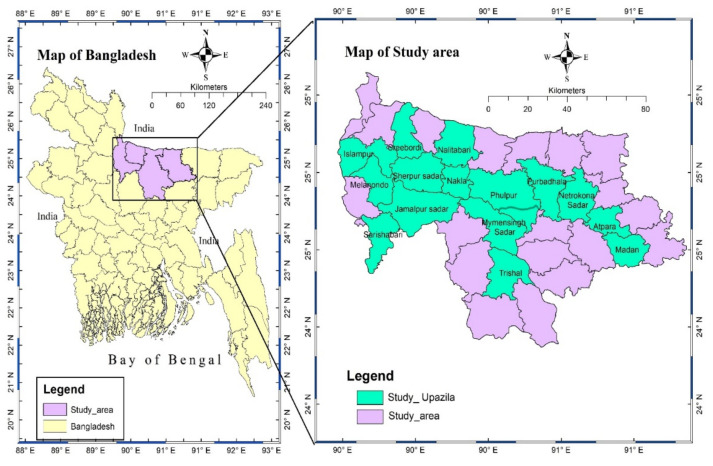
Maps of study area. Bangladesh is the country; Study_area means the Mymensingh division of Bangladesh; Study_Upazilla means the selected 16 upazillas within Mymensingh division of Bangladesh.

**Table 1 antibiotics-11-00442-t001:** Demographic and socio-economic characteristics of large-animal farmers (n = 212) in the study area.

Characteristics	Category	Frequency (Number)	Percentage (%)
District	Mymensingh	53	25
	Sherpur	52	24.5
	Jamalpur	54	25.5
	Netrokona	53	25
Sex	Male	182	85.8
	Female	30	14.2
Age	18–30 years	79	37.3
	31–40 years	71	33.5
	41–50 years	42	19.8
	>50 years	20	9.4
Education	Illiterate	43	20.3
	PSC	45	21.2
	JSC	22	10.4
	SSC	31	14.6
	HSC	41	19.3
	Graduate	20	9.4
	Masters	10	4.7
Training	Not received	110	51.9
	Received	102	48.1
Farm type	Dairy	96	45.3
	Buffalo	20	9.4
	Goat	48	22.6
	Sheep	16	7.5
	Beef Fattening	32	15.1
Farm population size (number of animals on individual farm)	3 to 5	45	21.2
	6 to 10	93	43.9
	11 to 20	48	22.6
	>20	26	12.3

PSC, primary school certificate; JSC, junior school certificate; SSC, secondary school certificate; HSC, higher school certificate.

**Table 2 antibiotics-11-00442-t002:** Test of statistical significance of variation in the respondents’ knowledge on AMU and AMR by their characteristics.

Characteristics	Category	Knowledge		Chi Square	Attitude		Chi Square	Practice		Chi Square
		Adequate	Inadequate	*p*-Value	Desirable	Undesirable	*p*-Value	Appropriate	In Appropriate	*p*-Value
District	Mymensingh	21 (39.6%)	32 (64.4%)	0.786	23 (43.4%)	30 (56.6%)	0.908	11 (20.8%)	42 (79.2%)	0.982
	Sherpur	24 (46.2%)	28 (53.8%)		24 (46.2%)	28 (53.8%)		12 (23.1%)	40 (76.9%)	
	Jamalpur	20 (37%)	34 (63%)		22 (40.7%)	32 (59.3%)		11 (20.4%)	43 (79.6%)	
	Netrokona	23 (43.4%)	30 (56.6%)		21 (39.6%)	32 (60.4%)		12 (22.6%)	41 (77.4%)	
Sex	Male	83 (45.6%)	99 (54.4%)	0.003	86 (47.3%)	96 (52.7%)	<0.001	40 (22%)	142 (78%)	0.808
	Female	5 (16.7%)	25 (83.3%)		4 (13.3%)	26 (86.7%)		6 (20.0%)	24 (80.0%)	
Age	18 to 30 years	33 (41.8%)	46 (58.2%)	0.413	32 (40.5%)	47 (59.5%)	0.682	11 (13.9%)	68 (86.1%)	0.044
	31 t0 40 years	34 (47.9%)	37 (52.1%)		34 (47.9%)	37 (52.1%)		23 (32.4%)	48 (67.6%)	
	41 to 50 years	15 (35.7%)	27 (64.3%)		17 (40.5%)	25 (59.5%)		9 (21.4%)	33 (78.6%)	
	>50 years	6 (30%)	14 (70%)		7 (35%)	13 (65%)		3 (15%)	17 (85%)	
Education	Illiterate	2 (4.7%)	41 (95.3%)	<0.001	8 (18.6%)	35 (81.4%)	<0.001	2 (4.7%)	41 (95.3%)	<0.001
	PSC	6 (13.3%)	39 (86.7%)		10 (22.2%)	35 (77.8%)		5 (1.1%)	40 (88.9%)	
	JSC	13 (69.1%)	9 (40.9%)		10 (45.5%)	12 (54.5%)		3 (13.6%)	19 (86.4%)	
	SSC	16 (51.6%)	15 (48.4%)		14 (45.2%)	17 (54.8%)		7 (22.6%)	19 (86.4%)	
	HSC	26 (63.4%)	15 (36.6%)		25 (61.0%)	16 (39%)		12 (29.3%)	29 (70.7%)	
	Graduate	17 (85%)	3 (15%)		14 (70%)	6 (30%)		10 (50%)	10 (50%)	
	Masters	8 (80%)	2 (20%)		9 (90%)	1 (10%)		7 (70%)	3 (3%)	
Training	Not received	19 (17.3%)	91 (82.7%)	<0.001	20 (18.2%)	90 (81.8%)	<0.001	3 (2.7%)	107 (97.3%)	<0.001
	Received	69 (67.6%)	33 (32.4%)		70 (68.6%)	32 (31.4%)		43 (42.2%)	59 (57.8%)	
Farm type	Dairy	43 (44.8%)	53 (55.2%)	0.065	46 (47.9%)	50 (52.1%)	0.028	26 (27.1%)	70 (72.9%)	0.065
	Buffalo	6 (30%)	14 (70%)		7 (35%)	13 (65%)		1 (5%)	19 (95%)	
	Goat	14 (29.2%)	34 (70.8%)		14 (29.2%)	34 (70.8%)		7 (14.6%)	41 (85.4%)	
	Sheep	6 (37.5%)	10 (62.10%)		4 (25%)	12 (75%)		2 (12.5%)	14 (87.5%)	
	Beef Fattening	19 (59.4%)	13 (40.6%)		19 (59.4%)	13 (40.6%)		10 (31.3%)	22 (68.8%)	
Farm size	3 to 5	9 (20%)	36 (80%)	<0.001	8 (17.8%)	37 (82.2%)	<0.001	2 (4.4%)	43 (95.6%)	<0.001
	6 to 10	29 (31.2%)	64 (68.8%)		31 (33.3%)	62 (66.7%)		7 (7.5%)	86 (92.5%)	
	11 to 20	28 (58.3%)	20 (41.7%)		27 (56.3%)	21 (43.8%)		19 (39.6%)	29 (60.4%)	
	>20	22 (84.6%)	4 (15.4%)		24 (92.3%)	2 (7.7%)		18 (69.2%)	8 (30.8%)	

PSC, primary school certificate; JSC, junior school certificate; SSC, secondary school certificate; HSC, higher school certificate; n = 212 respondents.

**Table 3 antibiotics-11-00442-t003:** Logistic regression analysis of the factors associated with respondents’ knowledge, attitudes, and practices of AMU and AMR.

Variable	Category	Knowledge			Attitude			Practice		
		Odds Ratio (Exp. B)	95% C.I		Odds Ratio (Exp. B)	95% C.I		Odds Ratio (Exp. B)	95% C.I	
			Lower	Higher		Lower	Higher		Lower	Higher
Sex	Female	1.000			1.000			1.000		
	Male	4.192	1.537	11.435	5.823	1.954	17.356	1.127	0.431	2.946
Age	18–30 years	1.000			1.000			1.000		
	31–40 years	1.281	0.672	2.443	1.350	0.707	2.578	2.962	1.320	6.645
	41–50 years	0.774	0.357	1.678	0.999	0.466	2.141	1.686	0.636	4.466
	>50 years	0.597	0.208	1.717	0.791	0.284	2.199	1.091	0.274	4.348
Education	Illiterate	1.000			1.000			1.000		
	PSC	1.123	0.288	4.369	1.250	0.441	3.54	2.562	0.470	13.981
	JSC	1.987	0.470	8.394	3.646	1.169	11.373	3.237	0.499	21.003
	SSC	3.231	0.835	12.496	3.603	1.268	10.236	5.979	1.148	31.141
	HSC	2.816	0.799	9.928	6.836	2.535	18.43	8.483	1.764	40.801
	Graduate	2.045	0.430	9.717	10.208	2.994	34.807	20.500	3.866	108.698
	Masters	2.513	0.401	15.746	39.375	4.345	356.834	47.833	6.734	339.766
Training	Not received	1.000			1.000			1.000		
	Received	10.014	5.252	19.094	9.844	5.19	18.67	25.994	7.730	87.414
Farm size	3 to 5	1.000			1.000			1.000		
	6 to 10	1.840	0.569	5.950	2.313	0.962	5.561	1.750	0.349	8.786
	11 to 20	2.515	0.623	10.157	5.946	2.292	15.43	14.086	3.046	65.134
	>20	23.147	4.214	127.131	55.500	10.848	283.951	48.375	9.344	250.451

PSC, primary school certificate; JSC, junior school certificate; SSC, secondary school certificate; HSC, higher school certificate; n = 212 respondents.

**Table 4 antibiotics-11-00442-t004:** Correlations among a farmer’s knowledge, attitudes, and practices towards AMU and AMR.

Correlations			Knowledge	Attitude	Practice
Spearman’s rho	Knowledge	Correlation Coefficient	1.000	0.593 **	0.393 **
		Sig. (2-tailed)	<0.001	<0.001	<0.001
		n	212	212	212
	Attitude	Correlation Coefficient	0.593 **	1.000	0.474 **
		Sig. (2-tailed)	<0.001	<0.001	<0.001
		n	212	212	212
	Practice	Correlation Coefficient	0.393 **	0.474 **	1.000
		Sig. (2-tailed)	<0.001	<0.001	<0.001
		n	212	212	212

** Correlation is significant at the 0.01 level (2-tailed), n = number of respondents.

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
