# Peer review of "A Survey on Knowledge, Attitude, and Practices of Large-Animal Farmers towards Antimicrobial Use, Resistance, and Residues in Mymensingh Division of Bangladesh"

_antibiotics, 2022, doi:10.3390/antibiotics11040442_

Round 1

Reviewer 1 Report

Comments to authors

This manuscript is reporting results from a survey conducted on farms of different animal species regarding the knowledge of producers on antimicrobial drug use and resistance. The study provides useful information for future interventions in Bangladesh to control the use of antimicrobial drugs in food animals.

My main concerns are:

 1) The study surveyed different animal species that most likely have different patterns of antibiotic use and different management practices. Therefore, the observed results could be attributed to the difference in production type which is a confounding factor for this study.

2) The result section must be shortened. Several results are presented in the text and in the tables and figures. Do not repeat in the text what is presented in the tables.

3) English proofreading is recommended for this manuscript. Please double-check the references provided in this manuscript.

4) I highly recommend authors include a section about the limitations of this study. A section on ethical considerations is needed.

Please see specific comments in the attached pdf.

Author Response

Reviewer#1

This manuscript is reporting results from a survey conducted on farms of different animal species regarding the knowledge of producers on antimicrobial drug use and resistance. The study provides useful information for future interventions in Bangladesh to control the use of antimicrobial drugs in food animals.

My main concerns are:

 1) The study surveyed different animal species that most likely have different patterns of antibiotic use and different management practices. Therefore, the observed results could be attributed to the difference in production type which is a confounding factor for this study.

Response: We appreciate your comment. As you indicated, we also agree with your point and accordingly we highlighted the production type and revised the manuscript.

2) The result section must be shortened. Several results are presented in the text and in the tables and figures. Do not repeat in the text what is presented in the tables.

Response: we agree with your comments and accordingly we removed the repetition of table, figure and text for same issue.

3) English proofreading is recommended for this manuscript. Please double-check the references provided in this manuscript.

Response: we revised our entire manuscript for possible English correction. In addition, we check our entire manuscript for possible correction of references.

4) I highly recommend authors include a section about the limitations of this study. A section on ethical considerations is needed.

Response: We appreciate your comment. According to your suggestions we added the limitations of this study and ethical considerations.

Reviewer 2 Report

This is a commendable study which has really great value in understanding what the educational need is of farmers is within Bangladesh. It is especially the discussion that is the highlight of this manuscript. The study is well written, although English editing is needed. The major concern is how the authors represent their outcomes, the tables and figures carry similar information with the added text highlights, this becomes repetitive. Also, the number of tables and figures are far too much. There is also a lot of statistics mentioned, and the reviewer questions if it all contributes to the quality of the journal. Otherwise, a good manuscript.

Keywords: The use of these abbreviations may not provide the widest attention to search words; suggesting the authors consider replacing as there are names that also refer to similar terms not covered by these abbreviations.

References

General inconsistency, please review the journal style again and follow the correct format.

Ref1, 26: incorrect format for website

Ref 9, 16, 21: Incomplete

Keywords: The use of these abbreviations may not provide the widest attention to search words; suggesting the authors consider replacing as there are names that also refer to similar terms not covered by these abbreviations.

Introduction

Line 67: English grammar

Line 71-75: Sentence too long, causing English grammar errors. Please spilt to improve clarity.

Line 78: Please replace “the wrong one” for clarity e.g.  inappropriate antimicrobial.

Line 78-80: Please rephrase for accuracy. Also, improve English grammar.

Line 82-82: Please correct English grammar

Introduction

It is suggested that the introduction is shortened. The main issues are explained in the last paragraphs, the issues on the first page can be significantly reduced so that the focus is with the aims and objectives of this study within Bangladesh

Methods:

Please indicate if the participants were well informed and that they could participate or withdraw at any time according to international ethics guidelines- consent aspects.

Results

The first table appear shortly after the heading, suggest that the text appear before the table and that the main issues are highlighted within the text.

Table 1 can be simplified. Is N=212, the last column can fall away if it is n(%). It also lacks footnotes: PSC, JSC, SSC and HSC? Also, Table 1 can be incorporated within Table 5 as the information are similar.

Line 162-163: English grammar

Line 166: Please replace 6 and 10 with six and ten

Line 169: These antibiotics are not spelled with capital letters with in text.

Figure1: It is a good figure, but urge the authors to try to make the percentage figures more visible, as it is currently very small and difficult to read. Please also improve the heading description to be less cryptic.

Line 186, 191, 197: English grammar

There is a repetitiveness on how the results are presented: Table 2 and Figure 2, Table 3 and Figure 3 and Table 4 and Figure 4; inclusive of the highlights within the text that accompanies these descriptions. I suggested that the authors relook at these presentations.  

Figure 2, 3 and 4 is an interesting presentation of the question outcomes. My concern is that you must read it intensely with the footnotes to make any sense of it. Is this the most optimal presentation for these outcomes?

Table 2, 3 and 4: Please clarify what units are used under these headings.

Table 5: Please indicate N.

Unfortunately the reviewer failed to follow all the statistics the authors tried to depict in section 2.7. If much of it already appear in the table, can this section not be simplified?

Supplementary

I am not convinced that the excel sheet contributes to the manuscript and suggest its removal. 

Author Response

Rewiever#2

This is a commendable study which has really great value in understanding what the educational need is of farmers is within Bangladesh. It is especially the discussion that is the highlight of this manuscript. The study is well written, although English editing is needed. The major concern is how the authors represent their outcomes, the tables and figures carry similar information with the added text highlights, this becomes repetitive. Also, the number of tables and figures are far too much. There is also a lot of statistics mentioned, and the reviewer questions if it all contributes to the quality of the journal. Otherwise, a good manuscript.

Response: According to the nature of the survey studies statistical analysis are the key for validation of results, therefore all sorts of statics are very much essential for this type of survey report.

Keywords: The use of these abbreviations may not provide the widest attention to search words; suggesting the authors consider replacing as there are names that also refer to similar terms not covered by these abbreviations.

Response: We appreciate your comment. Accordingly we revised the keywords to make it better understandable for reader as well as search.

References

General inconsistency, please review the journal style again and follow the correct format.

Ref1, 26: incorrect format for website

Ref 9, 16, 21: Incomplete

Response: we revised the references.

Keywords: The use of these abbreviations may not provide the widest attention to search words; suggesting the authors consider replacing as there are names that also refer to similar terms not covered by these abbreviations.

Response: We appreciate your comment. Accordingly we revised the keywords to make it better understandable for reader as well as search.

Introduction

Line 67: English grammar

Line 71-75: Sentence too long, causing English grammar errors. Please spilt to improve clarity.

Line 78: Please replace “the wrong one” for clarity e.g.  inappropriate antimicrobial.

Line 78-80: Please rephrase for accuracy. Also, improve English grammar.

Line 82-82: Please correct English grammar

Response: we revised the entire manuscript for possible errors.

Introduction

It is suggested that the introduction is shortened. The main issues are explained in the last paragraphs, the issues on the first page can be significantly reduced so that the focus is with the aims and objectives of this study within Bangladesh

Response: we revised the entire introduction section.

Methods:

Please indicate if the participants were well informed and that they could participate or withdraw at any time according to international ethics guidelines- consent aspects.

Response: This issue was well written in section 4.4. In addition, we also revised the section in revised manuscript again.

Results

The first table appear shortly after the heading, suggest that the text appear before the table and that the main issues are highlighted within the text.

Table 1 can be simplified. Is N=212, the last column can fall away if it is n(%). It also lacks footnotes: PSC, JSC, SSC and HSC? Also, Table 1 can be incorporated within Table 5 as the information are similar.

Line 162-163: English grammar

Line 166: Please replace 6 and 10 with six and ten

Line 169: These antibiotics are not spelled with capital letters with in text.

Figure1: It is a good figure, but urge the authors to try to make the percentage figures more visible, as it is currently very small and difficult to read. Please also improve the heading description to be less cryptic.

Line 186, 191, 197: English grammar

There is a repetitiveness on how the results are presented: Table 2 and Figure 2, Table 3 and Figure 3 and Table 4 and Figure 4; inclusive of the highlights within the text that accompanies these descriptions. I suggested that the authors relook at these presentations.  

Figure 2, 3 and 4 is an interesting presentation of the question outcomes. My concern is that you must read it intensely with the footnotes to make any sense of it. Is this the most optimal presentation for these outcomes?

Table 2, 3 and 4: Please clarify what units are used under these headings.

Table 5: Please indicate N.

Unfortunately the reviewer failed to follow all the statistics the authors tried to depict in section 2.7. If much of it already appear in the table, can this section not be simplified?

Response: We appreciate your comments. According to your all suggestions we revised our manuscript and make it track change mode for understanding our correction.

Supplementary

I am not convinced that the excel sheet contributes to the manuscript and suggest its removal. 

Response: We appreciate your comments. Accordingly we deleted supplementary material and XL file.

Round 2

Reviewer 1 Report

Thanks authors for addressing most of the comments. However, there are several comments are not addressed yet. Please look carefully at my comments in the attached pdf to improve the quality of the manuscript.

For example, the statistical analyses section is missing important information for any reader to evaluate the soundness of your results.

Please provide more detailed information on how you built your logistic regression models. Specify the variables that were categorized and the levels of categories for each variable. Did you check for collinearity? Did you check for interactions? How do you check the fitness of the final model?

You still have the map picture that is not cited in the text.

Table 2. Show the statistically significant differences among the levels of each variable using superscript letters.

Author Response

Please change "ORs" to "OR". It is odds ratio not odd ratios. Change throughout the manuscript.

Response: Corrected

Be consistent with the number of decimals in your numbers. Only one decimal after the decimal point is adequate. Please check throughout the manuscript.

Whenever you get p-value= 0.000 you should report it as "p<0.001

Response: Corrected

This is overall P-value. You must show the significant difference between each level. You can use superscript letters to differentiate between levels of variables

Response: if we add P-Values for each level would definitely make the reader boar. That’s why we would like to use overall P Value.

All figures and tables should be cited in the main text. This Figure is missing figure caption and not cited in the text. Also, it is confusing for me that you have study area and study upazila. This is confusing please clarify.

Response: Corrected and revised the section

Add reference for the software used here.

Response: added

What was the population size used to calculate the sample size. Add this information here

Response: added

Provide a copy of the paper-based questionnaire used in this study in supplementary materials

Response: the paper-based questionnaire used in this study is added in supplementary material

Provide reference for the acceptable values that you used

Response: added

Provide reference for value used.

Response: added

Please provide more detailed information on how you built your logistic regression models. Specify the variables that were categorized and levels of categories for each variable. Did you check for collinearity? Did you check for interactions. How you checked the fitness of final models.

Response: we revised the material section and added the necessary information.
